# Binary X-ray Sources in Massive Brans–Dicke Gravity

Grigoris Panotopoulos [1,*,†], Ángel Rincón [2,†] and Ilídio Lopes [3,†]

1 Departamento de Ciencias Físicas, Universidad de la Frontera, Casilla 54-D, Temuco 4811186, Chile
2 Sede Esmeralda, Universidad de Tarapacá, Avenida Luis Emilio Recabarren, Iquique 2477, Chile; aerinconr@academicos.uta.cl
3 Centro de Astrofísica e Gravitação-CENTRA, Instituto Superior Técnico-IST, Universidade de Lisboa-UL, Av. Rovisco Pais, 1049-001 Lisboa, Portugal; ilidio.lopes@tecnico.ulisboa.pt
* Correspondence: grigorios.panotopoulos@ufrontera.cl
† These authors contributed equally to this work.

**Abstract:** This study focuses on the X-ray emission of low-mass black hole binaries in massive Brans–Dicke gravity. First, we compute the accretion disk with the well-known Shakura–Sunyaev model for an optically thick, cool, and geometrically thin disk. Moreover, we assume that the gravitational field generated by the stellar-mass black hole is an analogue of the Schwarzschild space-time of Einstein's theory in massive Brans–Dicke gravity. We compute the most relevant quantities of interest, i.e., (i) the radial velocity, (ii) the energy and surface density, and (iii) the pressure as a function entirely of the radial coordinate. We also compute the soft spectral component of the X-ray emission produced by the disk. Furthermore, we investigate in detail how the mass of the scalar field modifies the properties of the binary as described by the more standard Schwarzschild solution.

**Keywords:** modified gravity; accretion disk models; X-ray astrophysics

## 1. Introduction

Einstein's general relativity (GR) is the first famous theory of gravity that beautifully explains well-known astrophysical and cosmological phenomena that are not explained by Newton's gravity. Despite its success, however, there are many other gravitational phenomena that GR does not explain. Notably, among others, dark matter and dark energy, the late-time acceleration of the Universe, the homogeneity problem, the baryon asymmetry, and the behavior of matter at extreme densities, such as in the cores of neutron stars [1].

The Brans–Dicke (BD) theory is another not so famous metric theory that, like GR, also agrees with observations. BD is widely used to build alternative cosmological models, black hole solutions, and interior solutions of compact relativistic stars. In the cosmological context, its success results from the fact that it is able to explain the inflationary epoch and the Universe's accelerating phase without invoking either exotic matter fields or dissipative processes [2]. As in improved black hole solutions and in pure scale-dependent gravity, where the gravitational coupling becomes a scale-dependent quantity, it is well-known that BD gravity is the simplest example of scalar-tensor theories of gravity where Newton's constant is treated as a dynamical scalar field. In a more general context, alternative theories of gravity are by now well accepted, and therefore, they have attracted considerable attention. Thus, in this work we study a straightforward generalization of the BD theory, known as massive Brans—Dicke theory of gravity [3,4].

Like many other GR generalizations, massive BD can now be tested against observations. This validation is now possible since experimental data coming from faraway sources have been obtained by astronomers using optical telescopes, gravitational wave detectors, and space missions. For example, the X-ray and $\gamma$-ray satellites, such as NASA's Chandra X-ray Observatory and ESA's XMM Newton, are providing high-precision spectroscopic studies of many astronomical sources.

Even though GR [5] has successfully passed many observational and experimental tests [6–8], alternative theories of gravity also pass those tests. They additionally provide well-motivated, good theoretical grounds for explaining dark matter, dark energy, renormalizability, etc. For reference, we can highlight the following ones: $f(R)$ theories of gravity [9,10], Hořava gravity [11,12], scale–dependent gravity [13–27], "improved" gravitational solutions [28–30], Weyl conformal gravity [31], and massive gravity [32]. Thus, as in BD gravity, improved black hole solutions, and pure scale–dependent gravity, the corresponding Newton coupling, treated as a scalar field, plays a prominent role by modifying the classical solutions. Our main goal in this article is the study of X-ray binaries within the massive BD theory of gravity [3,4]. In the present work we assume that the binary consists of a stellar-mass black hole (i.e., a mass of a few tens of solar masses) and a solar-like, low-mass companion star.

We start by recalling the following classical GR result: the static and spherically symmetric gravitational field generated by a point mass assuming a vanishing cosmological constant is given by the Schwarzschild geometry [33], and when allowing for a non-vanishing cosmological constant, the generated gravitational field is described by the Schwarzschild–(anti)-de Sitter space-time [34]. However, in alternative theories of gravity, in general, more sophisticated solutions with additional terms can be found. In the massive BD theory of gravity, in particular, it turns out that the metric tensor is still given by the Schwarzschild–de Sitter space-time, where now the effective cosmological term is identified to the mass of the scalar field; see Section 4. Here, we study the accretion process and the electromagnetic emission spectrum of X-ray binaries within this theory of gravity, assuming that the primary star of the binary is a black hole.

The work presented in this article is organized as follows: After this introduction, in the following two sections we summarize the observational characteristics of X-ray binaries, and the basic physics of an accretion disc in a binary. In Section 4, we discuss the binary's electromagnetic emission spectrum within massive Brans–Dicke gravity. Finally, we finish with some concluding remarks in the Section 5. We adopt the mostly positive metric signature $(-,+,+,+)$, and we work most of the time in geometrized units where $k_B = c = \hbar = 1 = G$, using the conversion rules 1 m = $5.068 \times 10^{15}$ GeV$^{-1}$, 1 kg = $5.61 \times 10^{26}$ GeV, and 1 K = $8.62 \times 10^{-14}$ GeV.

## 2. X-ray Binary Models

Scorpius X-1, being the first X-ray source discovered outside the Solar System [35], is the prototype of many high–energy binary systems. Each binary consists of a primary star, a compact object—either a black hole or a neutron star—and a companion star, a less massive star (donor) feeding an accretion disk around the compact star. The electromagnetic emissions of those binaries vary depending on the nature of the companion star, i.e., a main-sequence star, a red giant star, or a white dwarf star. Over the years, astronomers have discovered many other X-ray binaries. Consequently, we now have a good understanding of the fundamental fluid dynamics processes of binaries and their electromagnetic emission, including mass accretion and jet formation. In particular, we know that any binary produces an X-ray emission when the companion star transfers matter through the inner Lagrange point to the primary star, or the primary star captures mass from the companion star's wind. The mass transferred in a binary depends on: (a) the total amount of angular momentum; (b) the physical mechanism that regulates the loss of angular momentum; and (c) the radiation process that cools down the binary.

High-resolution observations of X-ray binaries realized by current space observatories provide one of the more compelling ways to test new theories of gravity. Indeed, the accretion of matter in binaries was studied originally in Newtonian gravity (see references [36–38]) and later generalized to curved space-times in [39]. In particular, isothermal Bondi-like accretion became a popular model on the galactic evolution community [40–42]. These processes have been investigated in the context of GR and alternative theories of gravity [43–68].

Of particular interest to us are the accretion processes occurring in the binary systems where a massive primary star evolves rapidly, ending quickly as a neutron star or a black hole. Additionally, in contrast, the companion star, being much lighter, stays on the main sequence strip burning hydrogen for a longer time. In a nutshell, the binary evolves as follows: the primary compact object captures matter coming from the Roche lobe of the companion star (the donor). As matter approaches the compact object, it starts to rotate in Keplerian orbits around the compact objects, leading to the formation of a disk. As the matter approaches, the accreting matter rotates faster and faster, heating up. The temperature eventually reaches millions of degrees Kelvin, and the binary starts to emit in the X-ray band of the electromagnetic spectrum.

Low-mass binary X-ray sources correspond to a binary system in which a primary star is a compact object—a neutron star or a black hole (in this work a stellar-mass black hole), and the companion star is a solar-like star in the main sequence. Those binaries are among the brightest extra-solar objects. The binary emits almost its entire energy in the X-ray band of the electromagnetic spectrum. The observed electromagnetic spectrum of such a binary has two components, namely, a soft and a hard one [69]. The latter is emitted from the surface of the primary star, and it corresponds to a black-body spectrum of $T \approx 2$ keV. In contrast, the former component, represented well by a multicolor black-body spectrum [69], comes from an optically thick accretion disk.

Binaries containing black holes possess X-ray emission spectra that depend on the properties of the black hole and the physics of the accretion disk. Therefore, the emission spectra of those binaries will also be influenced by the theory of gravity considered in the study. Moreover, as black holes are robust predictions of any metric theory of gravity, those binary X-ray sources are excellent cosmic laboratories to test new theories of gravity on the grounds of accretion disk modeling.

## 3. General Considerations

In what follows we shall briefly introduce the basic ingredients and the key assumptions required to compute the quantities of interest. In particular, we will summarize the details regarding how we effectively can define a suitable model for the accretion disk, and after that, we will discuss the physics related to accretion in spherical symmetric geometries. Subsequently, in the next section, we will present our main results.

### 3.1. Accretion Disk Model

The corresponding density is parameterized in terms of surface mass density $\Sigma(r)$, height $2H(r)$, and volume mass density $\rho(r)$ as follows

$$\rho(r) = \frac{\Sigma(r)}{2H(r)} \tag{1}$$

The properties of the disk depend on a considerable number of factors. To name a few, we should consider: (i) accretion rate, (ii) pressure, and (iii) opacity. An excellent review of the theory of black hole accretion disks may be found in [70]. In what follows, we adopt the standard model by Shakura–Sunyaev. This model, known since the 70s [71], is suitable for describing geometrically thin, optically thick and cool accretion disks. The main assumptions, summarized in [72], are the following:

1.    There are no magnetic fields.
2.    Advection is negligible.
3.    The accretion velocity, $u^i$, has a radial component $u^r < 0$ only; i.e.,

$$u^\mu = (u^0, u^r, 0, 0). \tag{2}$$

4.  The disk is geometrically thin, namely:

$$h(r) \equiv \frac{H(r)}{r} \ll 1. \tag{3}$$

5.  The disk is optically thick, as the opacity, $\kappa$, is dominated by the Thomson scattering:

$$\kappa = \sigma_T / m_p = 0.4 \, cm^2 / g, \tag{4}$$

where $m_p$ is the proton mass; $\sigma_T$ is the Thomson cross section; and the disc is opaque, characterized by a large optical depth, $\tau > 1$, and therefore, it is optically thick.

6.  The disk is cool: In geometrically thin, optically thick (Shakura–Sunyaev) accretion disks, radiation is extremely efficient, and nearly all of the heat generated within the disk is emitted (i.e., radiated) locally. Thus, the disk is considered to be cold; i.e.,

$$T \ll \frac{Mm_p}{r}, \tag{5}$$

where $M$ is the mass of the black hole and $T$ is the temperature of the disk. Additionally, notice that generically, when matter is optically thick $\tau > 1$, the accretion disk can be quite luminous and also (efficiently) cooled by radiation. Moreover, radiation is relevant in accretion disks as an efficient way to carry excess energy away from the system.

7.  The total pressure, $P$, has two contributions: the first one from the gas and the second one from radiation; i.e.,

$$P = P_{\text{gas}} + P_{\text{rad}} = \frac{T}{m_p} \rho + \frac{a}{3} T^4, \tag{6}$$

with $a = 4\sigma$ being the radiation constant, and $\sigma$ being the Stefan–Boltzmann constant.

8.  Accretion rate at the Eddingron limit: For a steady state disk, when a balance between gravity and pressure is reached, the accretion rate takes a constant value, $2\pi r \Sigma u^r = -\dot{M} = \text{constant}$, given by the Eddington limit [72]

$$\dot{M} = 16 L_{\text{Edd}} \tag{7}$$
$$L_{\text{Edd}} = 1.2 \times 10^{38} \left( \frac{M}{M_\odot} \right) \text{erg}/\text{s} \tag{8}$$

Although we are considering several assumptions, all of them are well justified and physically reasonable. The immediate consequence is that the problem may be described by a system of algebraic equations that is quite easy to solve.

In the Shakura–Sunyaev model, the temperature obeys the following profile [71]:

$$T(r) = \left[ \left( \frac{3M\dot{M}}{8\pi\sigma r^3} \right) \left( 1 - \sqrt{\frac{R_{in}}{r}} \right) \right]^{1/4} \tag{9}$$

where $R_{in}$ is the radius of the innermost stable circular orbit. When $r \gg R_{in}$, the temperature takes a simpler power-law form.

$$T(r) = \left( \frac{3M\dot{M}}{8\pi\sigma r^3} \right)^{1/4}; \tag{10}$$

i.e., it decays with the radial distance as $r^{-3/4}$. Notice that, for a stellar-mass black hole of a mass $M = (\text{tens}) M_\odot$, the temperature is $\sim (10^6 - 10^7) \, K$—see Figure 1 for $M = 30 \, M_\odot$. As $T \ll m_p$, the assumption for a low temperature disk is evidently satisfied.

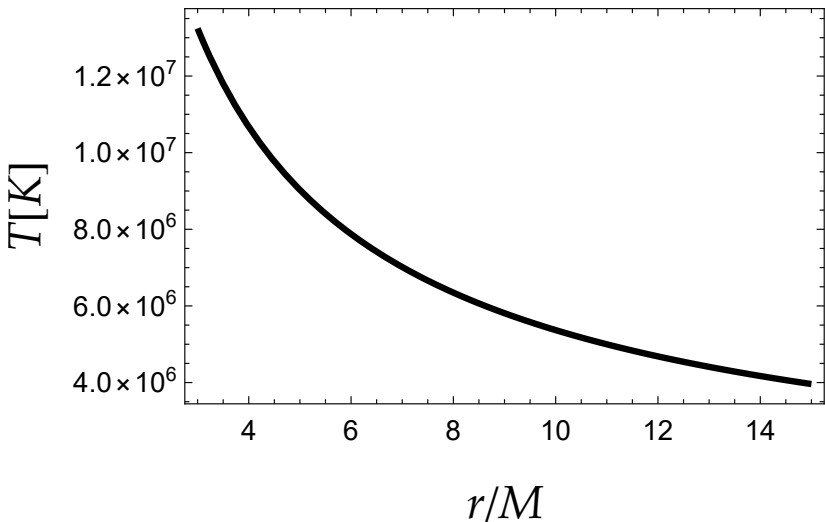

**Figure 1.** Temperature of the disk versus radial distance assuming $M = 30\ M_\odot$.

It is possible to compute the surface density and the semi-height, making use of (1) and the following Equation [72].

$$\Sigma = \frac{-\dot{M}}{2\pi r u^r} \tag{11}$$

Finally, the computation of the accretion velocity and the energy density will be presented in the next section. Besides, it should be mentioned that given the equation of state, the pressure may be computed once the temperature and the energy density are known.

### 3.2. Accretion in Spherically Symmetric Geometries

Theoretical black hole solutions admit a variety of effects which presently are considered unlikely to occur in nature. In such a sense, a (more) realistic black hole is usually considered electrically neutral, axisymmetric, Kerr-like [73], and endowed with several properties, such as: (i) its mass, $M$, and (ii) its rotation speed, $J$. The spin parameter $a^* = J/M^2$ (closely related to the rotation speed) in the very few known cases of high-mass X-ray binaries is found to be close to its extreme value, whereas in low-mass X-ray binaries (which is precisely the case investigated in the present work) covers the whole range, $0 \leq a^* \leq 1$ [74,75]. For simplicity reasons, as a first step we consider here non-rotating black holes. It certainly would be very interesting to investigate Kerr-like black holes in binaries as well, and we leave that study for a future work, where we expect to see deviations depending on the precise value of the spin parameter $a$. The deviation should be small for lower values of $a$, whereas higher values of $a$ should induce considerable deviations in comparison to the non-rotating black hole binary.

Therefore, in what follows we consider static, spherically symmetric space-times in Schwarzschild-like coordinates $t, r, \theta, \phi$—namely,

$$ds^2 = -f(r)dt^2 + f(r)^{-1}dr^2 + r^2 d\Omega^2 \tag{12}$$

with $f(r)$ being the corresponding lapse function.

Accretion processes for general spherically symmetric compact objects have been treated recently in [76], and after that, applications have appeared in the literature; see, for instance, [77–82] and references therein. Relativistic dust accretion onto a scale-dependent polytropic black hole [83] was analyzed in [84]. Additionally, in the context of massive gravity, there was a recent work by [85]. In the following we follow closely those two works [76,84,85].

For matter we assume a perfect fluid—namely,

$$T_{\mu\nu} = P g_{\mu\nu} + (\rho + P) u_\mu u_\nu \tag{13}$$

where the velocity of the fluid, $u^\mu$, satisfies the normalization condition

$$g_{\mu\nu} u^\mu u^\nu = -1 \tag{14}$$

or, more precisely

$$- f(u^0)^2 + (1/f)u^2 = -1 \tag{15}$$

In order to obtain a set of equations for $P, \rho$ and $u^0, u^r \equiv u$, we first use (i) an equation of conservation of energy and (ii) an equation of conservation of number of particles [72,84], i.e.,

$$\nabla_\mu(\rho u^\mu) = 0, \qquad \nabla_\nu T^{\mu\nu} = 0, \tag{16}$$

and using the last equations, we obtain:

$$\rho u r^2 = C_1 \tag{17}$$
$$(P + \rho) u^0 u r^2 = C_2 \tag{18}$$

with $C_1$ and $C_2$ being two arbitrary constants of integration. Notice that when $P \ll \rho$, it is easy to combine the equations to obtain an algebraic Equation (for the radial component), which can be written down as

$$u(r) = -\sqrt{C_3 f(r)^2 - f(r)} \tag{19}$$

where we have redefined $C_3 \equiv (C_2/C_1)^2$. As a final step, we go back to the original equations to obtain the energy density; i.e.,

$$\rho(r) = \frac{C_1}{u(r)r^2} \tag{20}$$

At this point it is evident that both the accretion velocity, $u^r = u$, and the energy density, $\rho$, can be obtained once the gravitational field, Equation (12), is specified. Additionally, as the pressure is negligible compared to the energy density, $P \ll \rho$, a concrete form of the equation of state has not been required up to now. Finally, according to the physical considerations regarding the properties of the accretion disc, we can now compute the pressure using the above equation of state, Equation (6). It is clear that there is an interplay between the theory of gravity (via the lapse function in the last equations) and the physics of the disk, Equations (6) and (10).

## 4. Stellar-Mass Black Hole X-ray Binary in Massive Brans–Dicke Gravity

Here we use the material presented in the previous section to compute the main properties of a binary X-ray source involving astrophysical black holes in alternative theories of gravity, in particular, in massive Brans–Dicke gravity. This theory is an example of a scalar-tensor theory, where the gravitational interaction is mediated by a scalar field. Within massive Brans–Dicke gravity, the action is given by [4]:

$$I[g_{\mu\nu}, \phi] \equiv \frac{1}{16\pi G} \int d^4x \sqrt{-g} \left[ \phi R - \frac{\omega}{\phi} \partial_a \phi \partial^a \phi - \frac{1}{2} m^2 \phi^2 \right] + I_M[g_{\mu\nu}, \phi] \tag{21}$$

where $G$ is the usual gravitational constant, $R$ is the scalar curvature of the metric tensor $g_{\mu\nu}$, $I_M$ is the action of matter fields, $\omega$ is the Brans–Dicke constant, and $m$ is a mass parameter. Here we have considered, for a massive BD theory, a potential of the form

$$U(\phi) \equiv \frac{1}{2}m^2\phi^2 \tag{22}$$

and for solar system tests, the important scale corresponding to a mass scale $m_{\mathrm{AU}} = 10^{-27}$ GeV [3]. Varying with respect to the metric tensor, $g_{\mu\nu}$, and the scalar field, $\phi$, one obtains the field equations as follows:

$$\phi\left(R_{\mu\nu} - \frac{1}{2}g_{\mu\nu}R\right) = 8\pi G T_{\mu\nu} + \frac{\omega}{\phi}\left(\partial_\mu\phi\partial_\nu\phi - \frac{1}{2}g_{\mu\nu}(\partial_a\phi)^2\right) + \nabla_\mu\partial_\nu\phi - g_{\mu\nu}\Box\phi - g_{\mu\nu}\frac{1}{2}U(\phi) \tag{23}$$

$$(2\omega + 3)\,\Box\phi = 8\pi G T + 2\phi\frac{\partial U}{\partial\phi} - 4U \tag{24}$$

where $T$ is the trace of the matter energy-momentum tensor.

The massless limit, $m = 0$, corresponds to the standard BD theory, and in addition, one recovers Einstein's GR in the limit $\omega \to \infty$. Besides, in the massive BD theory of gravity, the allowed region of the two-dimensional parameter space $\omega, m$ has been constrained in [3,4]. In the following, we assume for $m$ numerical values compatible with the results obtained there.

In what follows we consider Schwarzschild-like solutions (static and spherically symmetric without rotation), when $\phi = 1$ and for vanishing matter content. The metric function takes the simple form

$$f(r) = 1 - \frac{2M}{r} - \frac{1}{12}m^2r^2 \tag{25}$$

where $M$ is the black hole mass, and we have identified

$$2\Lambda \equiv \frac{6}{l^2} = \frac{1}{2}m^2 \tag{26}$$

Thus, one obtains a solution that coincides with the usual Schwarzschild–de Sitter geometry, although the effective cosmological constant is not the (tiny) one that accelerates the Universe, but it depends on the mass of the scalar field, $\Lambda = m^2/4$.

In Figure 2, we show both the pressure and the mass density as a function of the radial coordinate varying the mass parameter, $m$, assuming a stellar-mass black hole of mass $M = 30\,M_\odot$, and setting

$$C_1 = -10^{-13} \tag{27}$$
$$C_2 = -(5/2)C_1 \tag{28}$$

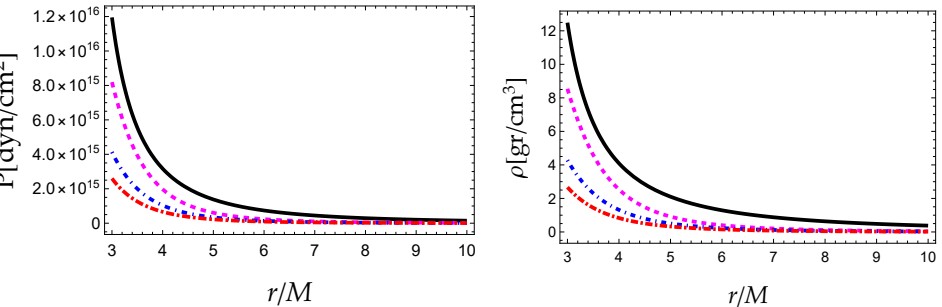

**Figure 2.** Pressure (**left**) and energy density (**right**) versus radial coordinate assuming $M = 30\,M_\odot$. The color code is given as follows: $m = 0$ (solid black line), $m = 40 \times 10^{-22}$ GeV (dashed magenta line), $m = 50 \times 10^{-22}$ GeV (short-dashed blue line), and $m = 60 \times 10^{-22}$ GeV (long-dashed red line).

It is easy to verify that $h(r) = $ constant is given by

$$h(r) = \frac{-\dot{M}}{4\pi C_1} \sim 10^{-7} \ll 1 \tag{29}$$

and therefore, one of the basic requirements is met. To conclude, the semi-height and also the surface mass density are given by

$$H(r) = 10^{-7}r \tag{30}$$
$$\Sigma(r) = 2 \times 10^{-7}\rho(r)r \tag{31}$$

The semi-height grows linearly with $r$, and the surface density is shown in Figure 3.

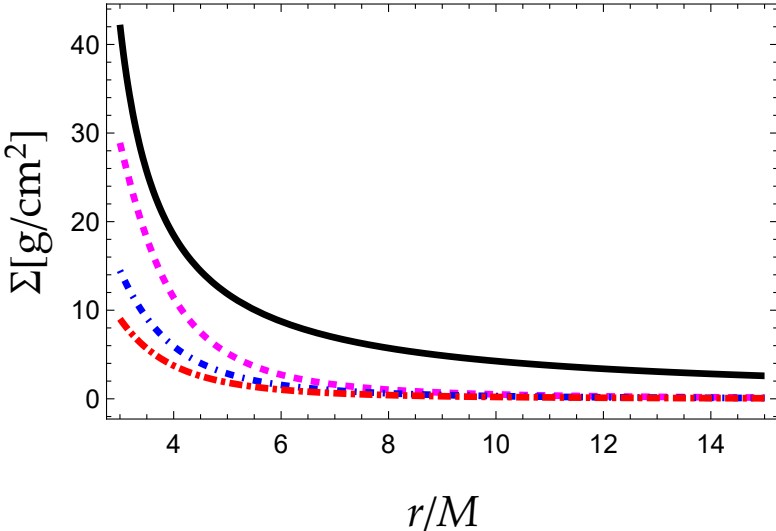

**Figure 3.** Surface density versus radial coordinate, $\Sigma(r)$, assuming $M = 30\ M_\odot$. The color code is given as follows: $m = 0$ (solid black line), $m = 40 \times 10^{-22}$ GeV (dashed magenta line), $m = 50 \times 10^{-22}$ GeV (short-dashed blue line), and $m = 60 \times 10^{-22}$ GeV (long-dashed red line).

*Flux of X-ray Emission*

At this point we switch to normal units. Thus, we write the numerical values of the constants taken from particle data group. The set of numerical values useful to us is: $M_\odot = 2 \times 10^{30}$ kg, $G = 6.67 \times 10^{-11}$ m$^3$/(kg s$^2$), $c = 3 \times 10^8$ m/s, $k_B = 1.38 \times 10^{-23}$ J/K, $\sigma = 5.67 \times 10^{-8}$ W/(m$^2$K$^4$), $h = 6.63 \times 10^{-34}$ gr, 1 pc $= 3.09 \times 10^{16}$ m, 1 erg $= 10^{-7}$ J, 1 eV $= 1.6 \times 10^{-19}$ J.

Regarding X-ray emission, the soft spectral component, $E\dot{F}_E$, expected from the optically thick disk, is obtained by means of the following expression [69].

$$F_E = \frac{1}{D^2}\ \cos(i)\ \int_{R_{\text{in}}}^{\infty} dr 2\pi r B(E, T) \tag{32}$$

where $B(E, T)$ is the Planckian distribution:

$$B(E, T) = \frac{2E^3}{c^2 h^3}\ \frac{1}{\exp[E/(k_B T)] - 1} \tag{33}$$

Here the numerical constants take the conventional values; i.e., (i) $c$ is the speed of light in vacuum, (ii) $h$ is the Planck constant, (iii) $D$ is the distance from the source, (iv) $k_B$ is

the Boltzmann constant, (v) $i$ is the inclination ($0°$ face-on, $90°$ edge-on), and (vi) $R_{\rm in} = 3r_H$. Making the change $s = r/R_{\rm in}$, the flux is computed by

$$F_E = \left( \frac{R_{in}}{D} \right)^2 \cos(i) \frac{4\pi}{c^2 h^3} E^3 \times \int_1^\infty ds \frac{s}{\exp[E/(k_B T(s))] - 1}, \tag{34}$$

or taking into account the spectral hardening factor, $f_{col}$, for a diluted blackbody spectrum [86–88], the soft X-ray emission energy flux is written as

$$F_E = \left( \frac{R_{in}}{D f_{col}^2} \right)^2 \cos(i) \frac{4\pi}{c^2 h^3} E^3 \times \int_1^\infty ds \frac{s}{\exp[E/(f_{col} k_B T(s))] - 1} \tag{35}$$

It should be mentioned that the impact of a given configuration (i.e., known distance, inclination, etc.) on the expected spectrum is revealed via a modification of $r_{\rm ISCO}$ of the black hole, as displayed in Figure 4 for a fictitious binary at distance $D = 10$ kpc, inclination $i = 70°$, and $M = 30\, M_\odot$ to simulate some of the binaries shown in Table 1 of [88]. The spectral hardening factor varies over a narrow range, $f_{col} = (1.4 - 2)$ [87], whereas in the sources shown in Table 1 of [88], it was found to be $f_{col} \approx 1.6$ in most of the cases. The computed spectrum as a function of the photon energy in a logarithmic plot exhibits the usual behavior with a peak at $E_* \sim$ (a few) keV.

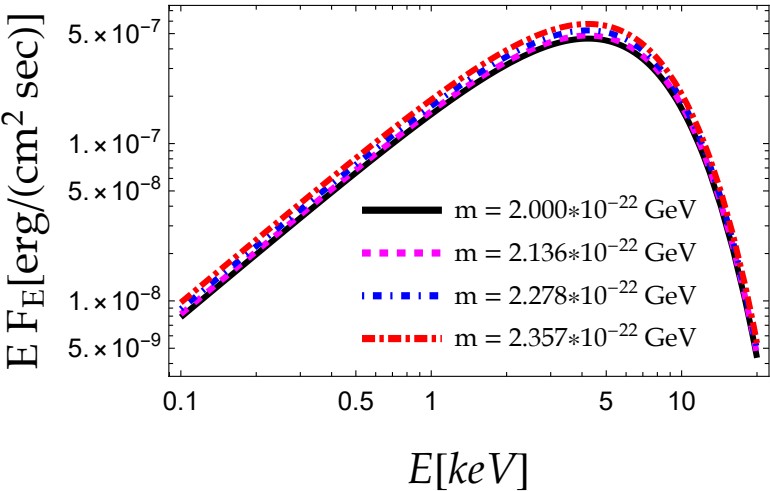

**Figure 4.** Soft X-ray emission from the black hole accretion disc assuming $D = 10$ kpc, $i = 70°$, $f_{col} = 1.6$ and $M = 30\, M_\odot$ for different values of the mass parameter $m$. The spectrum is slightly enhanced with $m$.

For the geometry considered here, the mass term ($\propto r^2$) is the key ingredient, and its inclusion shifts the curves upwards, maintaining the position of the peak. To obtain the modification with respect to the standard Schwarzschild case, we first obtain the $r_{\rm ISCO}$ by computing the roots of the effective potential $V_{\rm effec}$ (see, e.g., [89,90]) for different values of the parameter $l$ (or equivalently $m$). Subsequently, we perform the coefficient $z \equiv r_{\rm ISCO}/r_0$, where $r_0$ is the Schwarzschild radius in kilometers, adjusting the X-ray emission spectrum accordingly. Thus, for the numerical values considered here, the spectrum is enhanced in comparison to the standard case, $m = 0$. Therefore, X-ray astrophysics may in principle allow us to discriminate between Einstein's general relativity and the massive Brans–Dicke theory of gravity, although one should keep in mind that other non-standard theories of gravity lead to very similar spectra for binary X-ray sources; see, e.g., [85]. The challenge for the future would be to be able to disentangle the predictions of the Brans–Dicke theory of gravity from those of massive gravity. To that end, complementary observations may be needed.

In particular, the details on how to compute $r_{\text{ISCO}}$ can be found, for instance, in [89,90]. Avoiding the details, we need to find the lowest (among the real and positive) root of $P_4(r)$, a function found to be

$$P_4(r) = \sum_{i=0}^{4} b_i r^i \tag{36}$$

where the coefficients $b_i$ are computed to be

$$b_4 = -8 \tag{37}$$
$$b_3 = 30M \tag{38}$$
$$b_2 = 0 \tag{39}$$
$$b_1 = 2l^2 M \tag{40}$$
$$b_0 = -12l^2 M^2 \tag{41}$$

Although an analytic expression for the roots exists, it is so long that we prefer to avoid showing it. Instead, we obtained the roots numerically for $r_s = 88.8$ km, varying the numerical value of the parameter $l$. Thus, we assumed four different values of $l$ to compute $r_{\text{ISCO}}$. They were the following:

$$r_{\text{ISCO}}{}^{I} = 289.365 \text{ km} \text{ with } l = 3417.34 \text{ km} \tag{42}$$
$$r_{\text{ISCO}}{}^{II} = 295.889 \text{ km} \text{ with } l = 3200.00 \text{ km} \tag{43}$$
$$r_{\text{ISCO}}{}^{III} = 307.300 \text{ km} \text{ with } l = 3000.00 \text{ km} \tag{44}$$
$$r_{\text{ISCO}}{}^{IV} = 322.108 \text{ km} \text{ with } l = 2900.00 \text{ km} \tag{45}$$

As a final remark, let us mention that in both Brans–Dicke and Einsteinian gravity, a few simplifications are made. One of the most common assumptions is the Schwarzschild ansatz, which implies that $g_{rr} = g_{tt}^{-1}$. In the present work we have assumed that this condition holds, although in more complicated circumstances, the normalization condition used to obtain our results (the velocity profile) is therefore adjusted, making the problem more attractive. Thus, in light of the above comments, in alternative theories of gravity, the real impact of the assumption $g_{rr} \propto g_{tt}^{-1}$, or also, when the metric potentials are not related to each other, could be a quite interesting problem. This idea is left to be explored in future work.

The global behavior of our model is well characterized by the variations in the leading structure quantities: all four quantities, namely, temperature $T$, pressure $P$, energy density $\rho$, and surface density $\Sigma$ (see Figures 1–3) are monotonically decreasing functions of $r$, very similar qualitatively to the ones presented in [72]. The height $H(r)$ increases linearly with $r$, but the energy density decreases faster, and therefore overall the surface density also decreases with $r$. It is easy to verify that the assumptions of the non-relativistic standard model for geometrically thin, cold, and optically thick disks are met. The energy density depends on the background geometry and the lapse function $f(r)$, whereas the temperature depends on the details of the disk model only, and not on the background geometry. As far as the pressure is concerned, once $T$ and $\rho$ are known, it may be computed using the equation of state (6). Regarding the impact of the mass of the scalar field, the temperature is not affected, since as already mentioned, it does not depend on the geometry, whereas $P$, $\Sigma$, and $\rho$ decrease with $m$. Both the pressure and the surface density grow linearly with the energy density, as can be seen in Equations (1) and (6), respectively. Therefore, since $\rho$ decreases with the mass of the scalar field, $P$ and $\Sigma$ also decrease as $m$ increases.

## 5. Conclusions

In summary, we have studied the accretion disk and the soft spectral component of X-ray binaries within massive Brans–Dicke gravity. We have considered a binary system

consisting of a black hole (primary star) and a sun-like star in the main sequence (donor). Surrounding the black hole is a cool, optically thick, and geometrically thin accretion disk. We have modeled the accretion disk following the seminal paper by Shakura and Sunyaev, allowing us to describe the disk by a system of algebraic equations. In this model, the temperature decreases with the radial distance as $r^{-3/4}$. Following this, we have computed the pressure as a function of the radius by considering the two components of the disk plasma: gas and radiation.

Moreover, the gravitational field generated by the black hole is a static, spherically symmetric geometry (assuming a very low rotation speed) within Brans–Dicke massive gravity. In this model, we have found that the metric function $f(r)$ is given by the Schwarzscild–de Sitter geometry, where the affective cosmological constant is identified to the mass of the scalar field, $2\Lambda = 6/l^2 = m^2/2$. The ISCO radius depends on the parameter $m$; such that $r_{\text{ISCO}}$ increases with a decrease in $l$. Finally, the soft component of the X-ray spectrum emitted by the accretion disc corresponds to a superposition of multicolour blackbody spectra, i.e., a Planckian spectrum for each disk layer with a given temperature. We note that the emission spectrum depends on the geometry and on the ISCO radius for a fixed geometrical configuration (distance of the source, inclination, etc). Thus, we have shown graphically the new spectral component emitted from the disc resulting from massive BD gravity. We found that (i) the spectrum as a function of the photon energy in a logarithmic plot exhibits the usual behavior with a peak at $E_* \sim 5$ keV, (ii) it is shifted upwards compared to the spectrum corresponding to the standard Schwarzschild geometry, and (iii) for a given photon energy, it increases with the mass of the scalar field.

**Author Contributions:** Conceptualization, Á.R., G.P. and I.L.; methodology, Á.R., G.P. and I.L.; formal analysis, Á.R., G.P. and I.L.; investigation, Á.R., G.P. and I.L.; writing—original draft preparation, Á.R., G.P. and I.L.; visualization, Á.R., G.P. and I.L. All authors have read and agreed to the published version of the manuscript.

**Funding:** This research received no external funding.

**Institutional Review Board Statement:** Not applicable.

**Informed Consent Statement:** Not applicable.

**Acknowledgments:** The authors G. P. and I. Lopes thank the Fundação para a Ciência e Tecnologia (FCT), Portugal, for the financial support to the Center for Astrophysics and Gravitation-CENTRA, Instituto Superior Técnico, Universidade de Lisboa, through project UIDB/00099/2020 and grant number PTDC/FIS-AST/28920/2017. The author A. R. acknowledges the University of Tarapacá for support.

**Conflicts of Interest:** The authors declare no conflict of interest.

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
