# Peer review of "Binary X-ray Sources in Massive Brans–Dicke Gravity"

_universe, doi:10.3390/universe8050285_

Round 1

Reviewer 1 Report

Dear Editor and Authors, In this manuscript, the authors proposed to study the accretion disk
and the soft spectral component of X-ray binaries within massive
Brans-Dicke gravity. A binary system consisting of a black hole
(primary star) and a sun-like star in the main sequence is considered.
In this context, some quantities are calculated, such as the radial
velocity, the energy and surface density, the pressure as a function
of the radial coordinate, and the soft spectral component of the
X-ray emission produced by the disk. I think the analysis and
results presented in the manuscript are correct. They are interesting
and important for this area of research. The graphics are well
discussed. In my opinion, the manuscript is well written. Therefore,
I think the paper can be accepted for publication in Universe in its
current form.

Reviewer 2 Report

The discussed paper is devoted to the idea of looking for the difference between the massive Brance-Dicke (BD) theory and general relativity (GR). The authors check the accretion disk theory in GR and BD origins and present the possible differences that could be observed in real astronomy. The study looks interesting and having experimental consequences.

The only problem that seems to be not explained carefully is that in Section 3.1 and in the beginning of Section 3.2 the authors claim that the known black holes have a big angular momentum. The Shakura-Synaev theory of accretion discs also operate with Kerr metrics (see, for example, "Accretion flows in astrophysics" / N. I. Shakura, G. V. Lipunova, K. L. Malanchev et al. — New York: New York, 2018. — 419 p). But beginning from Equation (12) all the consideration is based on spherically-symmetric space-time without spinning.  Such consideration is a baby model and a first step? Or there are some physical reasons allowing such an approach? 

The next question is for the conclusions section: would the conclusions be the same if the spinning would be taken into account?

Please, provide more careful explanation before the Equation (12).

Author Response

This manuscript is a resubmission of an earlier submission. The following is a list of the peer review reports and author responses from that submission.

Round 1

Reviewer 1 Report

Comment to authors:

The authors study 'Binary X-ray source in massive Brans-Dicke gravity'. In this work, they examine the profiles of disk temperature, surface density, pressure, and mass density as a function of the radial coordinate. Moreover, they calculate the disk emission considering Shakura-Sunyaev (1973) model. The topic is interesting. However, I find that authors presented their work in a very casual manner. I find it difficult to understand the importance of their work as it is not coherently presented. There are four figures in the manuscript without any description and discussion. They started with a low mass X-ray binary source but presented results considering 30 M_Sun sources. Moreover, since their model solutions have a negligible effect on disk emission, it is not clear to me how their formalism marks new imprints for the development of the topic. Numerous scientific mistakes are present. For example, the surface of the primary source need not be responsible for the hard radiations, and treating white dwarfs as companion stars seems to be too optimistic. Further trouble builds due to poor English. Overall, this paper does not qualify for publication in its present form. Although the manuscript lacks the publication standard, however, I feel authors should get a chance to modify it. Accordingly, I suggest a major revision of the manuscript.

Author Response

Dear Editor,

We thank both reviewers for comments and suggestions. We have revised our
manuscript according to their suggestions. Our response is the following:

Reviewer 1: In the revised version of our manuscript, we have added a discussion
on the figures, and we have improved the English as much as we could. Regarding the other two points related to low-mass stars and white dwarf stars,
we wish to clarify that in our work we assume that the binary consist of a
stellar-mass black hole (i.e. of a mass of a few tens of solar masses) and a
solar-like star (rather than a white dwarf star). The latter is precisely the
low mass star.

Sincerely,

The authors

Reviewer 2 Report

The authors look at the x-ray observable (in particular, the thermal spectrum) from a binary system possessing a black hole described by massive Brans-Dicke gravity, which is an example of an alternative theory of gravity. The authors assume a spherically symmetric black hole and a geometrically-thin optically-thick disk around it. They assume the disk radiates as a blackbody and compare the radiation for different values of the mass parameter of the Brans-Dicke scalar field. 

The results make incremental progress in the larger program of tests of theories of gravity and therefore the manuscript deserves publication after correcting several grammatical errors that reduce readability.  

Author Response

Dear Editor,

We thank both reviewers for comments and suggestions. We have revised our
manuscript according to their suggestions. Our response is the following:

Reviewer 2: We wish to thank the reviewer for a positive report, and for pointing
out our grammatical mistakes. In the revised version of our manuscript we have
improved the English as much as we could.
Sincerely,

The authors

Round 2

Reviewer 1 Report

I do not find improvements as suggested. Authors failed to address the scientific merit of their work. The importance and critical implication of this work are not discussed. Overall, I find this work not suitable for publication at its present form.